# Smartphone Assessment of the Sitting Heel-Rise Test

**DOI:** 10.3390/s24186036

**Published:** 2024-09-18

**Authors:** Gustavo O. Hoffmann, Edilson Borba, Eduardo H. Casarotto, Gisele Francine Devetak, Ramzi Jaber, John G. Buckley, André L. F. Rodacki

**Affiliations:** 1Centro de Estudos do Comportamento Motor, Departamento de Educação Física, Setor de Ciências Biológicas, Universidade Federal do Paraná, Rua Coronel Heráclito dos Santos, 100, Centro Politécnico, Jardim das Américas, Curitiba 19011, Paraná, Brazil; gustavohoffmann@ufpr.br (G.O.H.); borba.edi@gmail.com (E.B.); henriqueeduardo@ufpr.br (E.H.C.); gidevetak@hotmail.com (G.F.D.); 2Faculty of Engineering & Informatics, University of Bradford, Bradford BD7 1DP, UK; r.jaber@bradford.ac.uk (R.J.); j.buckley@bradford.ac.uk (J.G.B.)

**Keywords:** gyroscope, heel-rise test, plantarflexors, smartphones

## Abstract

The study presents a new approach for assessing plantarflexor muscles’ function using a smartphone. The test involves performing repeated heel raises for 60 s while seated. The seated heel-rise test offers a simple method for assessing plantarflexor muscles’ function in those with severe balance impairment who are unable to complete tests performed while standing. The study aimed to showcase how gyroscopic data from a smartphone placed on the lower limb can be used to assess the test. Eight participants performed the seated heel-rise test with each limb. Gyroscope and 2D video analysis data (60 Hz) of limb motion were used to determine the number of cycles, the average rise (T-rise), lowering (T-lower), and cycle (T-total) times. The number of cycles detected matched exactly when the gyroscope and kinematic data were compared. There was good time domain agreement between gyroscopic and video data (T-rise = 0.0005 s, T-lower = 0.0013 s, and T-total = 0.0017 s). The 95% CI limits of agreement were small (T-total −0.1118, 0.1127 s, T-lower −0.1152, 0.1179 s, and T-total −0.0763, 0.0797 s). Results indicate that a smartphone placed on the thigh can successfully assess the seated heel-rise test. The seated heel-rise test offers an attractive alternative to test plantarflexor muscles’ functionality in those unable to perform tests in standing positions.

## 1. Introduction

Aging is a process characterized by significant changes in the neuromuscular system, which can compromise the ability to perform everyday tasks [1]. A reduced ability to produce and sustain force (i.e., strength and endurance, respectively) plays a central role in diminished physical performance, age-related functional impairments, frailty pathogenesis, and the risk of institutionalization, falls, and death [2,3]. Due to the diagnostic relevance of determining neuromuscular performance in older adults, there is a growing need for clinical assessments of muscle function that can be undertaken quickly and easily.

The strength and endurance of the triceps surae muscle are of particular interest, as the plantarflexor muscles play a crucial role in several quotidian tasks (e.g., dressing, bathing, step/stair ascent and descent, getting up from a chair, and walking) [1,2]. Among the various methods used to assess the strength and endurance of this muscle, the up-on-the-toes (UTT) test, also referred to as the heel-rise test [1] or calf-raise test [2], stands out for its practicality, low cost, and ease of application. This test involves repeatedly maximally raising the heels from the ground from a standing position as many times as possible within 30 s [4]. The number of repetitions has been used as a primary outcome to evaluate strength, endurance, fatigue, muscle function, and general performance [5,6,7].

Initially, the UTT test was proposed as a unilateral test, i.e., completed using a single limb [3]. Due to some older individuals having difficulties performing the test with a single limb, it was adapted into a bipedal test to investigate strength and balance deficits [8,9,10]. However, despite this adaptation, frail or infirm elderly participants often struggle with the bipedal test due to significant strength or functional deficits (e.g., weakness, unilateral surgery, or injuries) and/or sensory motor deficits (e.g., neurological or vestibular problems). In contrast, a comparable test performed in a sitting position (i.e., the seated heel-rise test) offers a more inclusive alternative, accommodating individuals with reduced balance or significant deficits in their ability to produce force. Furthermore, the seated heel-rise test can easily be used to assess the lower limbs individually, and this offers the opportunity to evaluate the symmetry between body sides. Thus, the seated heel-rise test may be a promising approach for assessing neuromuscular functioning in those with compromised balance or a marked deficit in producing sufficient force to raise their calves while standing (e.g., very old adults, Parkinson’s disease, chronic venous insufficiency, Achilles tendon injury, etc.). This practical, low-cost, and accessible test could be easily implemented across a variety of healthcare settings [11].

Recently, Zahid and colleagues used inertial measurement units (IMUs) to assess the UTT (standing) test performed by young adults and demonstrated an excellent agreement in the number of movements detected with that recorded by an observer [12]. Although their analysis successfully identified peak foot plantarflexion angular velocities across cycles, the approach did not consider temporal movement parameters (e.g., raising, lowering, and total cycle duration). Assessing temporal aspects of time series signals can offer more detailed insights into movement performance than focusing only on the number of repetitions. For instance, Lecumberri and colleagues reported that a set of kinematic parameters during the sit-to-stand test obtained from IMUs provided better frailty status identification than determining the number of cycles performed (i.e., the usual clinical outcome measure) [13]. Assessing the temporal parameters of the seated heel-rise test expands the analysis beyond the number of repetitions and might reveal other germane time domain features of the movement structure (e.g., the time spent in each phase). These aspects may help further explore several motor features (e.g., neuromuscular function, coordination, asymmetries, fatigability, etc.) in a broad context of clinical conditions (e.g., frailty, fall risk, sarcopenia, injuries, etc.). In addition, capturing the temporal aspects of seated heel-rise test movements using smartphone sensors is appealing due to the ubiquity of smartphones, which can facilitate the test’s application in various clinical and non-clinical environments (e.g., hospitals, clinics, health centers, etc.). Assessing time domain aspects in such a way can provide valuable insights into a patient’s functional capabilities and limitations, aiding more precise diagnostics, effective intervention planning, and monitoring clinical outcomes.

Therefore, this study aimed to present the seated heel-rise 60 s test as a new approach for assessing the function of the plantarflexor muscles in older adults who are incapable of completing the more traditional UTT test performed from a standing posture. It aimed to showcase how gyroscopic data obtained from a smartphone placed on the lower limb (distal thigh segment) can be used to quantify an individual’s performance in the seated heel-rise test, including the number of cycles completed along with the average and variability in rising, lowering, and total cycle durations.

## 2. Materials and Methods

### 2.1. Materials

Eight healthy older adults (3 males and 5 females; 75.5 ± 4.2 years, 1.60 ± 0.12 m, and 68.4 ± 11.3 kg) from the university’s local community and social projects volunteered to participate. Eligibility criteria included being 70 years or older, having no lower limb injuries or pain, and having no known vestibular or neurological conditions that could affect their performance in the test. Individuals with prostheses or recent surgeries were excluded. All participants met the inclusion criteria and provided written informed consent, and the study was approved by the Ethics Committee of the Paraná Federal University under the number 6.785.485.

### 2.2. Protocol

Participants attended a single 15 min laboratory visit. Initially, they provided personal information and basic descriptive anthropometrics and completed an anamnesis health questionnaire. Participants received verbal instructions and a demonstration of the protocol, with a practice attempt undertaken before starting the test.

During the test, participants sat with their backs against a chair and feet flat on the ground. The tests were performed barefoot and an adjustable support placed beneath the feet was adjusted to ensure the thigh was approximately parallel with the floor and the calf was perpendicular to the floor. A thin rubber mat was positioned beneath their feet to avoid slippage during the test. Participants were instructed to raise their heels as high and fast as possible, lightly touching the ground on each cycle. The test was performed for 60 s, but no information regarding the test duration was disclosed. Participants chose which limb was assessed first, with the contralateral limb assessed approximately 1 min later. During the test, participants rested one hand flat on their lap (palm facing down) and held a smartphone (iPhone 13 pro, Apple, model A2638, Cupertino, CA, USA) on the distal part of their thigh. No apparatus was used to secure the smartphone (i.e., tapes or strings), and the smartphone was maintained in place by requesting the participant to keep his/her hand flat on the top of the device. The smartphone was positioned along the medial line of the segment with the screen facing upward, and the projection of the upper edge of the device coincided with the superior aspect of the patella bone. The smartphone’s gyroscope data (x-axis) were collected using commercial software (Sensor Log, version 5.3) using a sampling frequency of 60 Hz.

The movements of the feet and lower limb were video recorded using a second smartphone (iPhone 13 pro, Apple, model A2638, CA, USA) placed on a tripod approximately 2 m away and perpendicular to the sagittal plane. Videos were recorded at 60 Hz with a resolution of 1920 × 1080 pixels. A 10 mm marker was drawn at the most prominent projection of the malleolus with a dermatographic pencil. After the seated heel-rise test for the first limb was recorded, the orientation of the chair the participant was seated on was reversed (turned 180 deg) to record the seated heel-rise test for the contralateral limb. Videos were processed frame-by-frame using Kinovea software (Version 2023.1.1). Figure 1 indicates the experimental setup.

Before each test, the experimenter performed a gentle tap on the phone within the camera’s visual field to serve as a reference for temporal synchronization between the video and smartphone sensors. The video information was assumed to represent the reference standard to which outcomes derived from the gyroscopic data were compared. The vertical time displacement of the ankle marker (y-axis) was obtained using the software’s automatic tracking tool, which negated any manual digitizing errors or variations and thus improved data reliability. The kinematic procedures used in the present study have been shown to have high intra- and interrater reliability (ICC > 0.98; [14]).

### 2.3. Signal Processing

Data obtained from the smartphone sensors presented small inconsistencies, i.e., the sampling frequency was inconsistent (ranging from 58 to 62 Hz). In addition, some of the data from the smartphone were assigned to the same time instant (i.e., two samples had the same timestamp). Thus, the time series data were resampled to a new time vector in which the frequency of 60 Hz was applied. This procedure ensured that the sampling frequency of the gyroscope data series matched the video data time series. The smartphone sensor’s convention was adjusted so that upward and downward movements were positive and negative, respectively. Finally, the video and gyroscope data series were filtered using the Savitzky–Golay filter using a window length of 31 points and a polynomial order of 4. The filtering procedure was deemed to have no prominent influence on the temporal aspects of the data series.

The ankle vertical displacement data (video) and the smartphone angular velocity data series (gyroscope) were time-synchronized so that the instant in which the first cycle was initiated was coincident in both data series. Then, the following instants were identified in the gyroscope data series: cycle initiation (INI—the zero-crossing instant before the peak positive angular velocity), maximum cycle vertical displacement (MAX—the zero-crossing instant after the peak positive angular velocity), and cycle termination (END—the zero-crossing after the peak negative angular velocity). The time between movement initiation (i.e., the instant heel rise begins) and maximum vertical displacement (i.e., the instant heel rise ends) was defined as the rising time (T-rise). The time between the maximum vertical displacement and the movement termination (the heel returns to the ground) was defined as the lowering time (T-lower). The total cycle time (T-total) was defined as the time between INI and END. In the video data series, INI was defined as the instant immediately prior to the ankle marker (y coordinate) raising from its lowest vertical displacement to initiate an upward trajectory toward its maximum vertical displacement. The MAX was defined as the instant of maximal amplitude in the y coordinate (“y peak”). The END was defined as the instant the y coordinate reaches its lowest value (i.e., the lowest displacement after the MAX).

All these instants were identified using a customized routine in which “find_peaks” (positive and negative) and “zero_crossings” functions were applied. The routines were written using free open-source software (Python, version 3.11.0). Figure 1 provides a representative sample of 3 consecutive cycles of the time domain aspects obtained from the video analysis and the gyroscope data series. The time domain instants and phases of interest are indicated. The video and gyroscope data series for a complete 60 s seated heel-rise test is also shown in Figure 1.

### 2.4. Statistics

To evaluate whether the analysis of the gyroscope data can provide an accurate/appropriate assessment of the seated heel-rise test, a Bland–Altman agreement analysis [15] was applied to the output parameters (i.e., the number of cycles, T-rise, T-lower, and T-total) determined using an analysis of the gyroscope data and those determined from the analysis of the video data. The Bland–Altman agreement for each outcome parameter was calculated considering data from all cycles performed by all participants (n = 1313 cycles). The statistical analyses were performed using JASP software (version 0.18.3).

## 3. Results

All participants were able to conclude the test and reported no discomfort or difficulties while performing the seated heel-rise test assessment. The number of cycles identified from the analysis of the gyroscope data series precisely matched those identified from the video data series analysis. On average, 82.1 ± 29.1 cycles were performed with each limb. However, there was a considerable variation in the number of cycles identified across participants, ranging from 43 to 131 repetitions. There was also variation between limbs in seated heel-rise test performance, with a mean variation of 10.3% in the number of cycles completed by one limb compared to the other. The number of cycles of each participant from the left and right segments and the percentage difference between the right and left segments are presented in Figure 2.

Table 1 provides the agreement in T-total, T-rise, and T-lower determined from the gyroscopic data with those determined using the video recording analyses. The Bland–Altman analyses indicated a good agreement in all phases of the seated heel-rise test (T-rise = 0.0005 s; CI95%: −0.0026, 0.0036 s; T-lower = 0.0013 s; CI95%: −0.0019, 0.0046 s; and T-total = 0.0017 s; CI95%: −0.0005, 0.0038 s, Figure 3).

## 4. Discussion

The current study presents the seated heel-rise 60 s test as a new approach for assessing the function of the plantarflexor muscles for older adults who are incapable of completing more traditional tests of plantarflexion function that are performed from standing. It showcases how gyroscopic data from a smartphone placed on the lower limb (distal end of the thigh segment) can be used to assess the seated heel-rise test. The number of seated heel-rise test cycles detected by the analysis of the gyroscope matched exactly the number of cycles determined by the video analysis, irrespective of the limb assessed and across all participants. In addition, there was a good agreement between time domain measurements obtained from the gyroscope data series and the video data series (agreement for T-total, T-rise, and T-lower, all within ≤0.0017 s).

### 4.1. Number of Cycles

The concordance between the number of cycles determined from the analysis of the gyroscope data series and the number determined by the video analysis aligns with comparable studies that have used a wearable device to analyze performance in the standing up-on-the-toes 30 s test (UTT-30, e.g., [12]). The average number of completed cycles is much larger (more than twice as large) than in studies that reported performance in the UTT-30 test (i.e., 82.1 ± 29.1 (in current study) vs. 11.8 repetitions—[4]). This is likely because of the lower muscle forces/demands required to raise the heel from the ground in a sitting position compared to a standing position, i.e., when raising the heels from the ground, the seated heel-rise test involves only elevation of the leg, whereas the UTT-30 test involves elevation of the whole body. On the other hand, the longer test period of the seated heel-rise test compared to the UTT-30 test (60 s vs. 30 s) may impose more pronounced effects on movement performance and thus may be more suitable to identify muscular endurance/fatiguing effects. Inter-subject muscle endurance differences might explain why the current study found a large range in the number of cycles completed across the different participants. While some participants were able to perform 131 cycles per minute, others performed at approximately one-third of this cadence (43 cycles per minute). It is important to note that no criteria about the physical status of the participants were used in recruitment, and thus we cannot retrospectively determine whether the physical status of participants was associated with the number of cycles they completed. The number of repetitions completed in a UTT-30 test has been advocated as a parameter to identify several clinical aspects, including decreased everyday mobility, plantarflexor endurance, and ankle strength deficits and asymmetries [9]. Future work is needed to determine if and how well performance on the seated heel-rise test is able to identify such clinical deficits.

Discrepancies in the number of cycles completed by the left and right limbs were also identified in the current study. Although a clinical interpretation of this finding is out of the study’s scope, functional differences between body sides have been related to postural control deficits [16], poor balance [17], gait asymmetry and variability [18], and an increased risk of falls [19,20,21]. Thus, the seated heel-rise test performance discrepancies between limbs may indicate critical functional differences but again, further investigation is required to confirm if the seated heel-rise test is able to identify such aspects.

### 4.2. Temporal Aspects

The temporal aspects of the seated heel-rise test cycles (time spent raising and lowering the heels from the ground and the total time to complete the cycles) determined by the analysis of the gyroscope data series had a good agreement with those determined from the video analysis. Previous studies looking at performance in the UTT-30 test aimed to control for repetitions not “fully” completed using several different means, including use of an electrogoniometer [7], height-adjustable elastic bands [22], foam touch-boards [23], and a plantarflexion angular velocity threshold measured via IMUs [12]. In the current study, we did not attempt to control for repetitions not completed fully because it was speculated that performance in raising and lowering the heels from the ground might decline over the 60 s, and this decline in performance was something that we thought could be relevant to assess. Although several previous UTT-30 testing protocols were based on a threshold performance (i.e., raising the heels as fast and high as possible) in which a certain output had to be sustained (e.g., a certain height threshold or pace must be sustained throughout the test [10,23]), little attention was paid to other relevant variables of the movement. Analyzing how temporal characteristics of repeated movements change over time is relevant, as such features may reflect whether the neuromuscular system is prone to the influence of several mechanisms (e.g., fatigue, control deficits, asymmetry, etc.) that may reduce the participants’ ability to sustain a temporal structure of the movement consistently. These changes could include having the rising and lowering times and consequently, cycle duration increasing toward the end of the test as a consequence of the natural fatigue process that occurs in response to successive repetitions. These changes in the temporal parameters over the time of the test may constitute an attractive proxy for assessing muscular performance and other parameters, such as fatigability, weakness, neuromuscular control issues, joint health problems, balance, stability concerns, and functionality. However, once again, further research is needed to explore and confirm such arguments.

The seated heel-rise test total cycle time assessed from video and gyroscope data showed high agreement, characterized by a small discrepancy in the mean total cycle time determined by each measurement modality. The associated small 95% limits of agreement reinforce that the gyroscope data series provides reliable estimates of the cycle duration when compared to the video analysis. On the other hand, the agreement in the estimates for the T-rise and T-lower times were not quite as good. These slightly increased measurement discrepancies may have arisen from difficulties in determining the precise instant of when the peak vertical velocity of the heel occurred in a particular cycle. In some cases, it was challenging to establish an unambiguous peak velocity instant, as small oscillations at the peak vertical displacement (i.e., the transition between the end of the rising phase and the beginning of the lowering phases) introduced some uncertainty. In addition, the fast transition between phases introduced some movement artifacts (wobbling, vibration, or fast alternating changes in positions of small amplitude) and precluded an unequivocal identification of the transition between the rising and lowering phases. Additionally, the smartphone was held in position by the participant’s bare hand, which may have increased the uncertainty sources due to differences in how the smartphone was held. Despite such sources of uncertainty, the agreement between the raising and lowering times was high (0.056 and 0.058, respectively), indicating that gyroscope measures were comparable to those obtained from the video analysis. It is possible to infer that the speed at which the tests were performed did not influence the results, as the errors are randomly distributed in fast and slow movements (Figure 3).

Because smartphones are ubiquitous, a positive aspect of the present study is related to the use of such devices for assessing the seated heel-rise test. Asking the participants to hold the smartphone with their hands on the distal portion of the thigh segment meant that no additional apparatus was necessary to perform the test. This highlights that the approach presented can be undertaken quickly and simply. However, it should be noted that holding a smartphone at the distal portion of the thigh may be challenging for patients with neurological conditions (e.g., Parkinson’s, stroke patients). Using apparatus to secure the smartphone (satchels, Velcro straps, tapes, etc.) may help to reduce movement artifacts but would place an additional burden that limits the ease and simplicity of the test application. Finally, although the video analysis was assumed to be a reliable reference standard for comparison with the gyroscopic determined movement outcomes, it is known that video analysis is not completely exempt from errors [24]. Future studies could identify the height of each cycle and then determine the instant the ability to sustain, within a certain percentage of the average height, starts to decline. Such a measure may constitute an attractive proxy for the functional status of the plantarflexor muscles. It is also relevant that future studies should identify the potential influence of the hip flexor muscles in helping raise the thigh segment from the chair seat during the test.

## 5. Conclusions

The results highlight that a gyroscope data series from a smartphone positioned on the thigh of a participant can be used to assess the number of cycles completed and the time domain features of performing a seated 60 s heel-rise test. Future work is needed to determine if performance in the seated heel-rise test is related to actual plantarflexor performance (e.g., strength and endurance). If the seated heel-rise test relates to strength and endurance, it may constitute an attractive alternative to test plantarflexor muscles’ functionality in those incapable of performing the more traditional clinical tests of plantarflexor muscle function that are executed from a standing position.

## Figures and Tables

**Figure 1 sensors-24-06036-f001:**
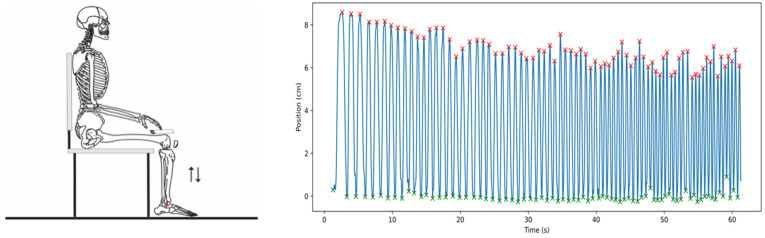
Schematic representation of the seated heel-rise test (**upper left** panel), an indication of the time domain aspects to define the cycle instants (initiation—INI; maximal displacement, MAX; termination, END) and phases (rising—T-rise; lowering—T-lower; total time—T-total) (**lower left** panel), as well as the video (**upper right**) and gyroscope (**lower right**) data series.

**Figure 2 sensors-24-06036-f002:**
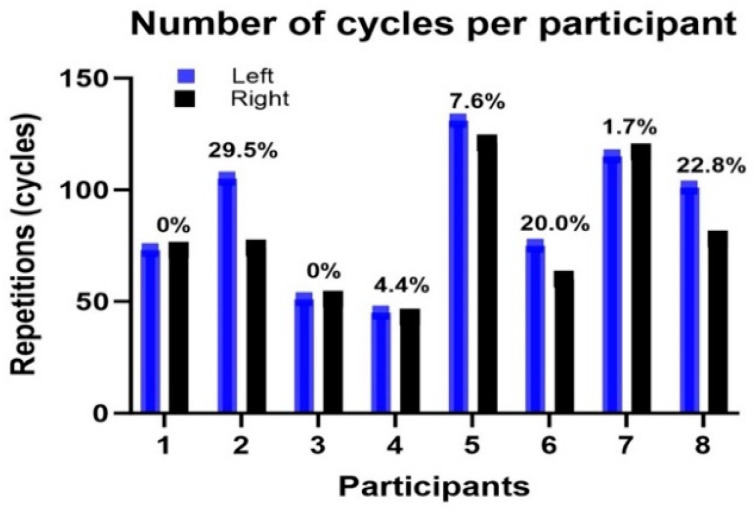
Number of cycles of each participant’s left and right sides during the seated heel-rise test. Note: the percentages indicate the intra-subject inter-limb variation (i.e., between the left and right sides).

**Figure 3 sensors-24-06036-f003:**
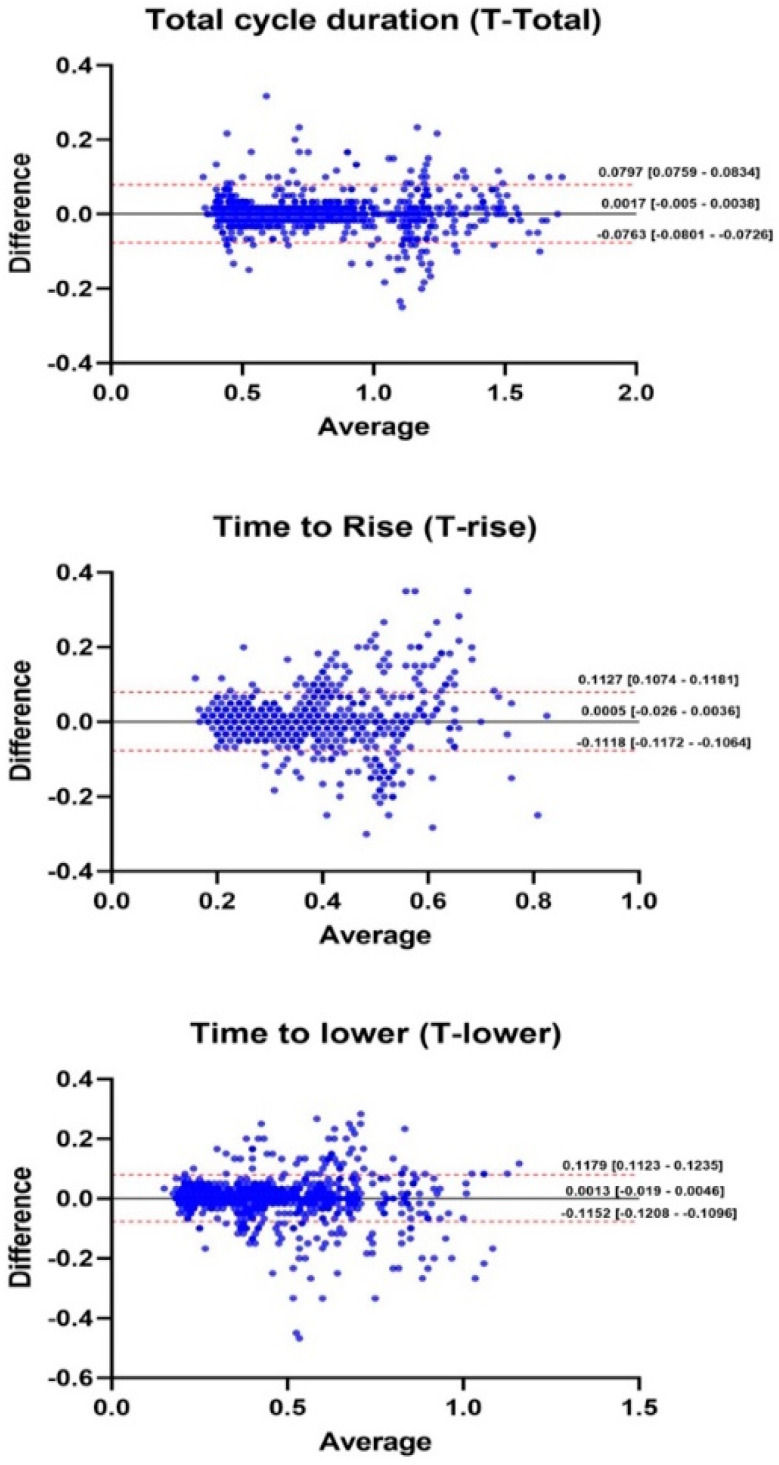
Bland-Altman plots considering all cycles performed by all participants (n = 1313 cycles) during the seated heel-rise test.

**Table 1 sensors-24-06036-t001:** The number of cycles, time to rise (T-rise), time to lower (T-lower), and total cycle time (T-total) determined from the video compared to the gyroscopic data compared to an analysis of video data and the respective Bland-Altman agreement and 95% confidence intervals.

	Data Series	Mean (±SD)	Agreement	95% CI
Cycles (number)	Gyroscope/ Video	82.1 (29.1) *	0.00	0, 0
T-rise (s)	Gyroscope	0.359 ± 0.116	0.0005	−0.1118, 0.1127
	Video	0.360 ± 0.113		
T-lower (s)	Gyroscope	0.454 ± 0.193	0.0013	−0.1152, 0.1179
	Video	0.455 ± 0.190		
T-total (s)	Gyroscope	0.814 ± 0.299	0.0017	−0.0763, 0.0797
	Video	0.815 ± 0.297		

* The number of cycles identified from the gyroscope and video were identical.

## Data Availability

The data supporting reported results can be made available upon reasonable request.

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
