# Peer review of "Smartphone Assessment of the Sitting Heel-Rise Test"

_sensors, 2024, doi:10.3390/s24186036_

Round 1

Reviewer 1 Report

Comments and Suggestions for Authors

Dear Dr Hoffmann and colleagues,

It was a pleasure to review your manuscript. I have provided a number of suggestions below:

Introduction

1.      Line 38: I do not think most readers would recognise the ‘up on the toes test’. This seems to be most commonly referred to as a calf raise test or heel raise test.  

2.      Line 44: I would suggest clarifying that the single leg test is only a problem for some individuals, many people can do it fine.

3.      Unless there is a problem with the word count, please spell out SHR and other abbreviations to avoid confusion for the reader given they are not common terms.

4.      It may be valuable to include some discussion on other calf raise smartphone applications:

o   Fernandez MR et al. Concurrent validity and reliability of a mobile iOS application used to assess calf raise test kinematics. Musculoskelet Sci Pract. 2023 Feb;63:102711. doi: 10.1016/j.msksp.2022.102711. Epub 2022 Dec 20. PMID: 36604270.

o   Hébert-Losier K et al. A randomised crossover trial on the effects of foot starting position on calf raise test outcomes: Position does matter. Foot (Edinb). 2024 Jun 17;60:102112. doi: 10.1016/j.foot.2024.102112. Epub ahead of print. PMID: 38905944.

5.      Line 57: Whilst this study may have found excellent agreement, other recent research has shown that technical cues can significantly alter results:

o   Green, B et al. Form Matters—Technical Cues in the Single Leg Heel Raise to Failure Test Significantly Change the Outcome: A Study of Convergent Validity in Australian Football Players. BioMed 2024, 4, 89-99.

Methods

6.      Please amend ‘men and women’ to ‘males and female’ as I assume these are in reference to sex, and not gender.

7.      Line 93: Please add the university details who provided the ethical approval.

Results

8.      Line 183: In there a typo on this line? It does not seem to make sense.  

9.      Figure 2 and 3: Can you amend the word ‘cycles’ to seated calf raises or something to make it clear for the reader.

10.  Table 2: Please amend the line relating to the cycles so it is clear both gyroscope and video reported the same number of cycles.

Discussion

11.  Line 236: I do not think you can report mobility is a potential problem when your test did not go into maximal dorsiflexion.

12.  Given the number of repetitions, I think it most appropriate you use the terminology endurance, and not strength throughout the paper

13.  Line 256: Some discussion about whether a score should be used for when the participants height starts to decline, and then the total repetitions until failure as it may be the important value is how many repetitions can be used until the height is lost. Could you explore the agreement on the cut off point for when rep height was reduced?

Limitations

14.   Please add information here on the different amount of plantarflexion between people, and that performing the raises off a step, to allow more depth by dropping into dorsiflexion might be worthwhile.

Author Response

Dear Reviewer:

On behalf of all authors, we appreciate your time and efforts in improving the manuscript's quality. Please find attached the detailed responses we addressed using a point-by-point approach. We have also included the responses to the other reviewer so you may have a complete idea of the points we have adjusted.

Many Thanks

André - corresponding author

Reviewer 2 Report

Comments and Suggestions for Authors

This paper developed an approach for assessing plantarflexor muscles’ function using a smartphone. There are some questions for the authors: 

1. How the smartphone was placed on the thigh? Just put on the thigh or fixed on the thigh by tapes or strings? .  

2. Since every subject has different speed in doing the SHR task,does this influence the obtained results?

3. Is the smartphone location the same for all the subjects? Is there any requirement of the smartphone direction?

4. Why the plantarflexor muscles’ function can be assessed by the cycles in SHR task? Since the number of cycles may depend on many factors.

5. What are the applications of the proposed methods ?

6. What are the strengths and limitations of this study? Please add corresponding illustration.

Author Response

(The authors gave the same response as above.)

Round 2

Reviewer 2 Report

Comments and Suggestions for Authors

The authors have answered my questions well.